# Poly-glutamine-dependent self-association as a potential mechanism for regulation of androgen receptor activity

Carlos M. Roggero[1,2,3]*, Victoria Esser[1,2,3], Lingling Duan[4], Allyson M. Rice[1], Shihong Ma[5], Ganesh V. Raj[5], Michael K. Rosen[1,6], Zhi-Ping Liu[4,7], Josep Rizo[1,2,3]*

**1** Department of Biophysics, University of Texas Southwestern Medical Center, Dallas, Texas, United States of America, **2** Department of Biochemistry, University of Texas Southwestern Medical Center, Dallas, Texas, United States of America, **3** Department of Pharmacology, University of Texas Southwestern Medical Center, Dallas, Texas, United States of America, **4** Department of Internal Medicine, University of Texas Southwestern Medical Center, Dallas, Texas, United States of America, **5** Department of Urology, University of Texas Southwestern Medical Center, Dallas, Texas, United States of America, **6** Howard Hughes Medical Institute, University of Texas Southwestern Medical Center, Dallas, Texas, United States of America, **7** Department of Molecular Biology, University of Texas Southwestern Medical Center, Dallas, Texas, United States of America

* Carlos.Roggero@UTSouthwestern.edu (CMR); Jose.Rizo-Rey@UTSouthwestern.edu (JR)

**Data Availability Statement:** All relevant data are within the manuscript and its Supporting Information files.

## Abstract

The androgen receptor (AR) plays a central role in prostate cancer. Development of castration resistant prostate cancer (CRPC) requires androgen-independent activation of AR, which involves its large N-terminal domain (NTD) and entails extensive epigenetic changes depending in part on histone lysine demethylases (KDMs) that interact with AR. The AR-NTD is rich in low-complexity sequences, including a polyQ repeat. Longer polyQ sequences were reported to decrease transcriptional activity and to protect against prostate cancer, although they can lead to muscular atrophy. However, the molecular mechanisms underlying these observations are unclear. Using NMR spectroscopy, here we identify weak interactions between the AR-NTD and the KDM4A catalytic domain, and between the AR ligand-binding domain and a central KDM4A region that also contains low-complexity sequences. We also show that the AR-NTD can undergo liquid-liquid phase separation in vitro, with longer polyQ sequences phase separating more readily. Moreover, longer polyQ sequences hinder nuclear localization in the absence of hormone and increase the propensity for formation of AR-containing puncta in the nucleus of cells treated with dihydrotestosterone. These results lead us to hypothesize that polyQ-dependent liquid-liquid phase separation may provide a mechanism to decrease the transcriptional activity of AR, potentially opening new opportunities to design effective therapies against CRPC and muscular atrophy.

## Introduction

Prostate cancer is one of the most common forms of cancer in men [1]. Survival and proliferation of prostate cancer cells depend critically on signaling through the androgen receptor (AR)

**Funding:** J.R. RP120717-P3 Cancer Prevention and Research Institute of Texas (CPRIT) https://www.cprit.state.tx.us/ J.R. I-1304 Welch Foundation https://welch1.org/ Z.P.L. RP120717-AC and RP120717-P1 Cancer Prevention and Research Institute of Texas (CPRIT) https://www.cprit.state.tx.us/ Z.P.L. RO1 CA215063 NIH https://www.nih.gov/ The funders had no role in study design, data collection and analysis, decision to publish, or preparation of the manuscript.

**Competing interests:** The authors have declared that no competing interests exist.

and, consequently, standard therapies against prostate cancer involve androgen reduction through chemical or surgical castration [2, 3]. Unfortunately, prostate cancer cells eventually develop the ability to activate AR independently of androgen and patients invariably develop the more aggressive castration-resistant prostate cancer (CRPC), which is associated with a much worse prognosis [4–6]. Hence, understanding the mechanisms underlying AR function and how AR is activated in a ligand-independent manner is crucial to develop effective therapies against CRPC.

One aspect of AR biology that is most likely involved in prostate cancer is the complicated functional interplay between AR and epigenetic enzymes, including histone deacetylases (HDACs) and lysine demethylases (KDMs). Thus, AR activity can be downregulated or upregulated by HDACs and KDMs (e.g. [7–11]), and the development and progression of prostate cancer is accompanied by extensive abnormalities in the levels of these enzymes and in histone modification marks [12, 13], which has been referred to as an 'epigenetic catastrophe' [14]. Moreover, AR was reported to bind to several KDMs, including KDM4A, although only limited information was reported about the regions of the KDMs and of AR implicated in the interactions [9–11]. Hence, the nature of these interactions remains unclear and it is unknown how they affect AR activity.

AR contains a long N-terminal domain (NTD), a central DNA-binding domain (DBD), a short hinge region (H) and a ligand-binding domain (LBD) (Fig 1A). The LBD is normally key for receptor activation, but the NTD is required for full transcriptional activity, and interactions between NTD and LBD regulate androgen-dependent gene expression [15–17]. Activation involves the translocation of AR to the nucleus when androgen binding to the LBD induces a large conformational change that exposes a nuclear localization signal (NLS) present between the DBD and LBD [18]. The NTD has attracted much attention not only because of its known participation in transcriptional activity but also, and in particular, because diverse C-terminally truncated splice variants of AR lacking the LBD have been identified in prostate cancer cell lines and in clinical specimens [19]. These variants can explain the development of androgen-independent AR activity in CRPC. Indeed, multiple studies have suggested that AR-V7, which is the most abundant of these variants and is constitutively localized to the nucleus, mediates androgen-independent cell growth and resistance to androgen deprivation (reviewed in ref. [20]). Interestingly, our recent finding that KDM4B regulates the generation of AR-V7 by alternative splicing upon androgen deprivation [21] provided another connection between AR and KDM function.

The AR NTD includes sequences that are collectively named activation function 1 (AF1) and are important for AR-dependent gene regulation, interacting with the general transcription factor TFIIF and co-activators such SRC-1, CBP and ART27 (reviewed in [22]). The NTD contains abundant low-complexity sequences, which are compositionally biased regions that are common in transactivation domains of transcription factors. In the AR NTD, these low complexity sequences include polyQ and poly-G regions (S1A Fig) with variable length in the human population [23]. This variability arises because of slippage of DNA polymerase on the DNA template containing multiple copies of the same codon during DNA replication [18]. Exceedingly long polyQ sequences (> 38 Qs) can lead to muscular atrophy [24], which may be associated to the formation of oligomeric fibrils [25]. Conversely, short polyQ length (< 20 Qs) was shown to increase the risk for prostate cancer [26, 27], indicating that longer polyQ sequences may have a protective effect. These findings may arise because the transcriptional potency of AR is inversely correlated with polyQ length [28–30].

As expected because of the abundance of low complexity sequences, much of the AR NTD is predicted to be intrinsically disordered (S1B Fig). This prediction has been supported experimentally [31], but some sequences do have defined propensities to form secondary structure,

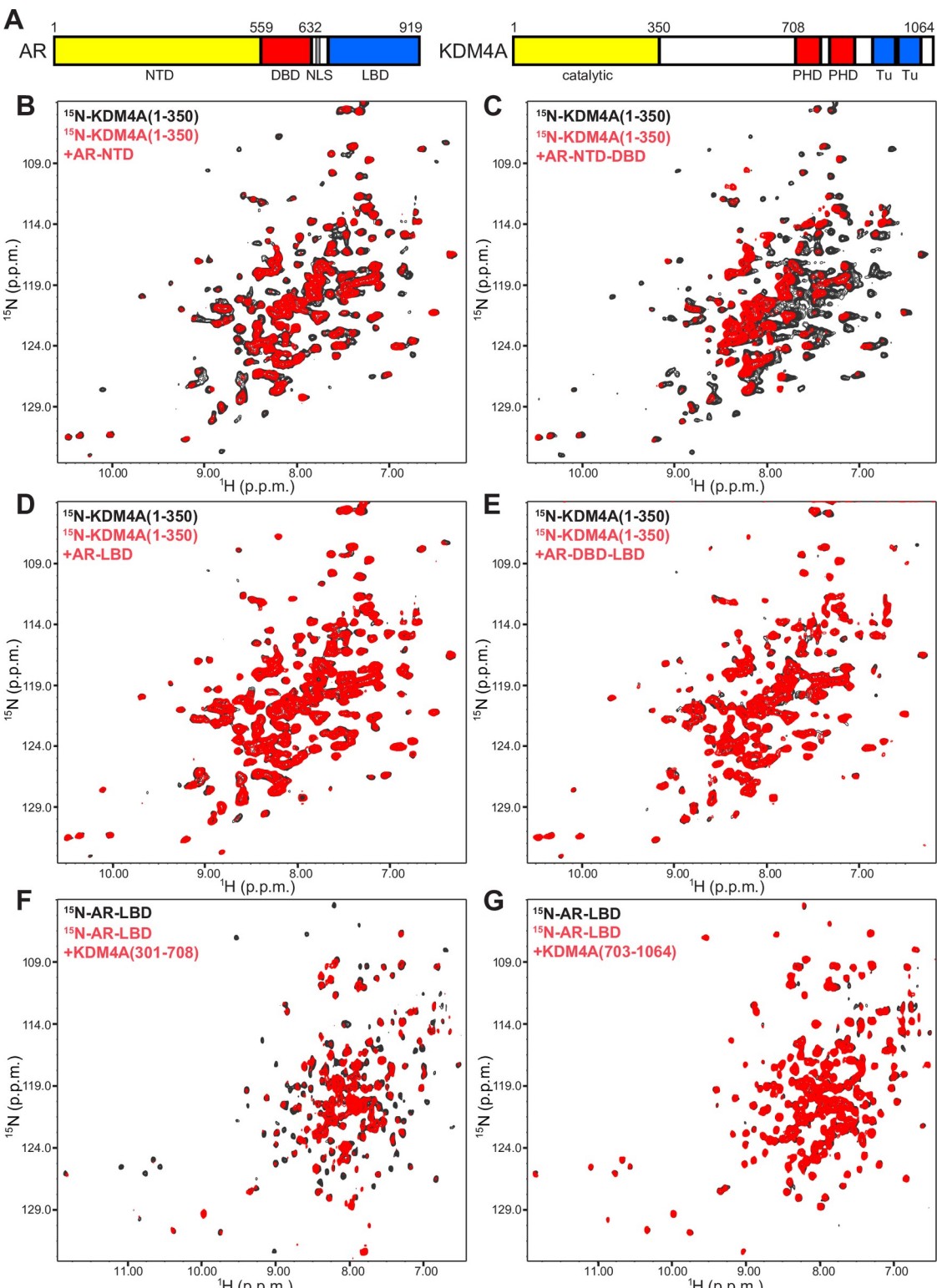

**Fig 1. Low complexity sequences are involved in binding between AR and KDM4A.** (A) Domain diagrams of AR and KDM4A. Selected residue numbers indicating the protein N- or C-termini, or approximate domain boundaries, are indicated above the diagrams. Domain names are shown below the diagrams (Tu = Tudor). (B-E) Superpositions of $^1$H-$^{15}$N TROSY-HSQC spectra of $^{15}$N-KDM4A(1–350) alone (black contours) or in the presence of equimolar amounts of unlabeled AR-NTD (B), AR-NTD-DBD (C), AR-LBD (D) or AR-DBD-LBD (E). (F, G) Superpositions of $^1$H-$^{15}$N TROSY-HSQC spectra of $^{15}$N-AR-LBD alone (black contours) or in the presence (red contours) of equimolar amounts of KDM4A(301–708) or KDM4A(703–1064).

particularly in regions that are important for transactivation [32, 33]. The polyQ sequence has a clear tendency to form α-helical structure that increases with the length of the sequence and is stabilized by interactions of the glutamine side chains with the backbone [34–36]. These observations led to the proposal that the effects of polyQ length on transcriptional activity may arise from changes in the strength of protein-protein interactions, for instance those with transcriptional co-regulators or general transcription factors [35]. It is also important to note that the AR NTD has been shown to undergo liquid-liquid phase separation at high concentrations (100 μM) [37], and that a propensity for phase separation appears to be a fundamental property of many low-complexity sequences that are involved in transcriptional activation and/or mediate formation of membraneless organelles [38–40]. However, the importance of this property for AR function and the influence of polyQ length on the tendency of the AR NTD to undergo phase separation have not been explored.

The study presented here was designed to examine how AR binds to KDM4A and to investigate how self-association of the AR NTD and nuclear localization of AR depend on the length of the polyQ sequence. NMR experiments reveal weak interactions between the AR-NTD and the KDM4A catalytic domain, and between the AR LBD and a central KDM4A region that also contains low-complexity sequences. We also show that the drive for the AR-NTD to undergo phase separation in vitro increases progressively with polyQ length. We verify that polyQ length is inversely correlated with transcriptional activity in two prostate cancer cell lines. Moreover, we find that nuclear localization is decreased by longer polyQ sequences in the absence of hormone, and that such sequences increase the formation of AR-containing puncta in nuclei of cells treated with dihydrotestosterone (DHT). Based on these results, we propose that longer polyQ sequences in AR may have a protective role against prostate cancer by enhancing self-association processes that reduce the AR transcriptional activity.

## Results

### Direct AR-KDM4A binding through regions containing low-complexity sequences

The present study was initiated with the goal of gaining insights into how KDMs affect AR function by investigating the nature of the interactions between these proteins. KDM4A and its homologues contain an N-terminal catalytic domain, a central region that contains low-complexity sequences predicted to be intrinsically disordered (S1C Fig), and a C-terminal region that includes two PHD domains and two Tudor domains, which recognize methylated basic residues [41] (Fig 1A). Co-immunoprecipitation experiments suggested that AR binds to KDM4A and KDM4D through its LBD, while the catalytic domain and the C-terminus of KDM4A, or the C-terminus of KDM4D, mediate AR binding [11]. However, it is difficult to derive definitive conclusions with regard to direct physical interactions from co-immunoprecipitation. To examine whether we could observe direct binding between purified recombinant proteins, we performed extensive pulldown assays and hold up assays [42] using fragments that spanned different regions of AR and KDM4A. However, we were unable to obtain definitive results using these approaches, suggesting that potential interactions between AR and KDM4A are too weak to be reliably detected by these methods.

We turned to an NMR method based on the analysis of perturbations caused by an unlabeled protein on transverse relaxation optimized (TROSY) $^1$H-$^{15}$N heteronuclear single quantum coherence (HSQC) spectra of a uniformly $^{15}$N-labeled protein, which provide a sensitive tool to detect interactions between the two proteins [43]. In a first set of experiments, we acquired $^1$H-$^{15}$N TROSY-HSQC spectra of a $^{15}$N-labeled fragment spanning the catalytic domain of KDM4A (residues 1–350) in the absence and presence of different unlabeled

fragments of AR. DHT was included in experiments with fragments containing the AR LBD, which is unstable without a ligand. A fragment spanning the NTD of AR (AR-NTD; residues 1–559) caused noticeable but limited broadening on the $^1$H-$^{15}$N TROSY-HSQC spectrum of $^{15}$N-KDM4A(1–350), while a longer fragment that included the AR NTD and DBD (AR-NTD-DBD; residues 1–632) caused stronger broadening (Fig 1B and 1C). In contrast, almost no perturbations on the $^1$H-$^{15}$N TROSY-HSQC spectrum of $^{15}$N-KDM4A(1–350) were caused by fragments containing the AR LBD alone (AR-LBD; residues 663–919) or from the DBD to the LBD (AR-DBD-LBD; residues 554–919) (Fig 1D and 1E). These results show that, under these conditions, the catalytic domain of KDM4A binds weakly to AR-NTD and that such binding is enhanced by the DBD, while there is no appreciable binding to the LBD or to the DBD in the absence of the NTD. Hence, binding of the AR-NTD-DBD fragment to KDM4A(1–350) appears to be driven by weak interactions with the NTD that can cooperate with additional interactions involving the DBD. The latter might be too weak to be observable without such cooperation or might be hindered by intramolecular binding of the DBD to the hinge region or the LBD in the context of the AR-DBD-LBD fragment, as outlined in the model of S2A Fig.

Because the AR-LBD had been previously implicated in binding to KDM4s [11] but we did not observe binding to KDM4A(1–350), we tested whether the LBD might interact with other regions of KDM4A. For this purpose, we prepared $^{15}$N-labeled AR-LBD bound to DHT and acquired $^1$H-$^{15}$N TROSY-HSQC spectra in the absence and presence of fragments spanning the central region of KDM4A (residues 301–708) or its C-terminal region containing the PHD and Tudor domains (residues 703–1064). The central region of KDM4A containing the low complexity sequences caused substantial broadening on the $^1$H-$^{15}$N TROSY-HSQC spectrum of $^{15}$N-AR-LBD, indicative of binding, while the C-terminal KDM4A fragment induced very little perturbations (Fig 1F and 1G).

We also analyzed whether the AR-NTD-DBD fragment binds to the catalytic domains of two other KDM4 isoforms, KDM4B and KDM4C. The $^1$H-$^{15}$N TROSY-HSQC spectrum of $^{15}$N-labeled catalytic domain of KDM4C exhibited dramatic broadening upon addition of AR-NTD-DBD, while the broadening was less overt in the $^1$H-$^{15}$N TROSY-HSQC spectrum of $^{15}$N-labeled catalytic domain of KDM4B (S2B Fig). Hence, there is some level of specificity in the interaction of AR-NTD-DBD with KDM4 isoforms.

Overall, our results indicate that there are indeed direct interactions between AR and KDM4A. The interactions appear to be weak, as they were difficult to detect using pulldown assays, but may cooperate with other interactions among components of the transcriptional machinery for recruitment of these proteins and/or control of their activities. Interestingly, the two types of interactions that we observed involve regions containing low complexity sequences of AR (binding of AR-NTD to the KDM4A catalytic domain) or KDM4A (binding of the KDM4A central region to AR-LBD).

## Longer polyQ sequences enhance gel formation by the AR-NTD

Increasing experimental evidence shows that many proteins that are multivalent for protein-protein interactions have a tendency to undergo liquid-liquid phase separation and that this property underlies the formation of membraneless subcellular compartments that serve a wide variety of biological functions [39]. In such systems, the drive to phase separate increases with protein length and/or interaction valence [39, 44]. The tendency to phase separate is particularly common for low-complexity sequences present in transcription factors and other proteins involved in formation of complexes with nucleic acids (e.g. [38, 40]). Indeed, the AR-NTD (residues 1–559) was shown to phase separate

into liquid droplets at 100 µM concentration [37]. These findings, together with the known importance of the AR-NTD for development of CRPC, the notion that longer polyQ sequences in the AR-NTD may protect against prostate cancer, and our finding that low complexity sequences mediate AR-KDM4A interactions, led us to investigate how polyQ length affects the ability of AR-NTD to phase separate. For this purpose, we prepared and purified recombinant fragments corresponding to the AR-NTD that contained polyQ sequences with 12, 20, 31 and 49 glutamines (AR-NTD-12Q, -20Q, -31Q and -49Q, respectively).

The four versions of AR-NTD each exhibited some turbidity at 500–600 µM concentrations and 200 mM NaCl, consistent with the previous observation of phase separation by this AR fragment [37], but the turbidity was markedly stronger for the AR-NTD fragments with longer polyQ sequence (Fig 2A). After a sample of AR-NTD-31Q was allowed to reach high turbidity, the sample became clear upon addition of 10% 1,6-hexanediol (Fig 2B). This observation shows that phase separation by NTD is reversible and does not result from irreversible formation of amyloid aggregates [45]. To have a more quantitative measure of the influence of polyQ length on the drive to phase separate, we prepared different dilutions of the four AR-NTD fragments from freshly prepared concentrated solutions, incubated the resulting samples for 1 h at 4°C, room temperature (RT; ca. 23°C) or 37°C, and measured the absorption at 400 nm. Analysis by SDS PAGE illustrated the purity of the samples used in these experiments (S3 Fig). Precipitation prevented reliable quantification of the turbidity at 37°C, but consistent results were obtained in the experiments performed at 4°C and RT, which generally revealed progressive increases of absorption at 400 nm with the concentration for the four proteins. Importantly, the turbidity correlated with the length of the polyQ sequence under the majority of conditions (Fig 2C and 2D; note that the lower OD at 400 nm observed for 100 µM AR-NTD-49Q after 1 h at RT, compared to that observed for 50 µM AR-NTD-49Q or 100 µM AR-NTD-31Q, might arise because droplets may settle out of solution). These results indicate that longer polyQ sequences increase the drive for the AR-NTD to phase separate.

The length of the polyQ sequence in AR was found to be inversely correlated with transcriptional activity [28–30]. To verify these findings with the same polyQ tracts that we used for the turbidity assays, we used a luciferase assay in two prostate cancer cell lines (LNCaP and PC3) transfected with vectors expressing full-length AR that contained 12, 20, 31 and 49 glutamines (AR-12Q, -20Q, -31Q and -49Q, respectively). In LNCaP cells, which express endogenous AR, we observed robust enhancement of transcriptional activity upon addition of DHT even when cells were transfected with control vector (Fig 3A). The DHT-induced increase in activity was markedly stronger for cells transfected with AR-12Q, but the stimulation gradually decreased as the polyQ length increased, and there was no additional stimulation in cells expressing AR-49Q (Fig 3A). To ensure that these results did not arise from lower expression of the AR variants with longer polyQ sequences, we analyzed the total AR levels in Western blots with AR antibodies and the levels of transfected AR proteins, which included a FLAG tag, with FLAG antibodies (Fig 3B). These experiments showed that the expression levels of the transfected AR proteins were comparable to each other, and similar to the levels of endogenous AR. In experiments performed with PC3 cells, which do not express endogenous AR, we did not observe a substantial DHT-induced enhancement of transcriptional activity when the cells were transfected with control vector, but transfection with AR-12Q caused robust DHT-dependent increase in activity and such increase again decreased gradually with the length of the polyQ sequence (Fig 3C). Hence, these results confirm that the length of the polyQ sequence can indeed have strong effects on AR activity.

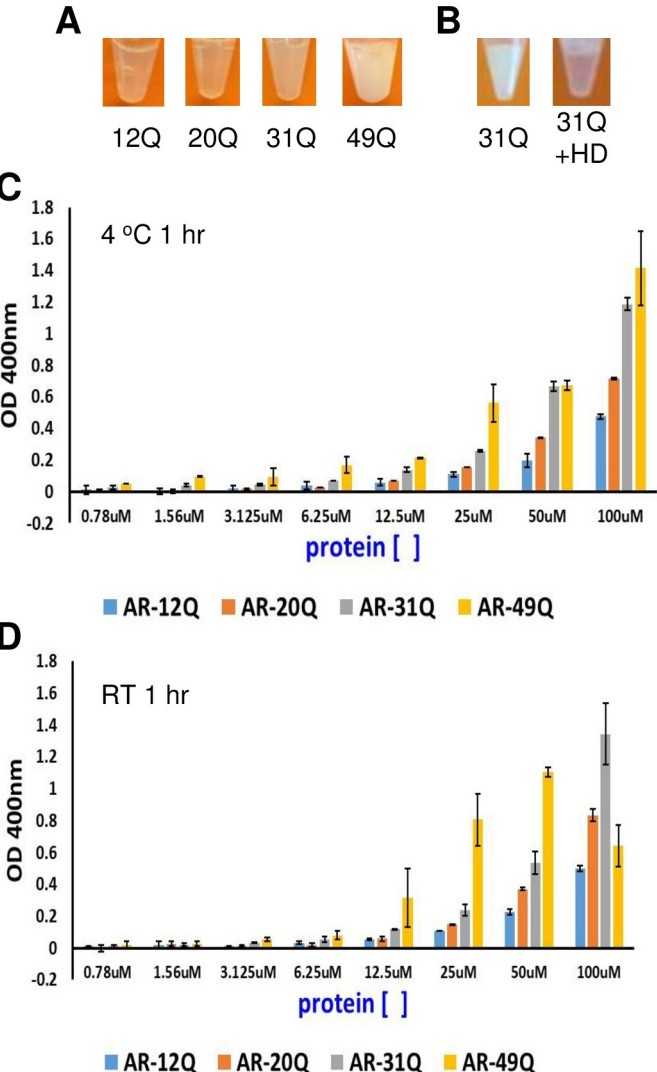

**Fig 2. Longer polyQ sequences increase the tendency of AR-NTD to form gels.** (A) Samples containing AR-NTD with different polyQ length were concentrated to 600 µM (12Q), 520 µM (20Q), 520 µM (31Q) and 480 µM (49Q) and incubated at RT for 24–48 h until gel formation was observed. Images of the resulting samples are shown. (B) Images of a 500 µM sample of AR-NTD-31Q that was allowed to form gels for 48 h before (left) and after (right) adding 10% 1,6-hexanediol (HD). (C, D) Turbidity assays with samples of AR-NTD containing different polyQ length. Samples were concentrated to 100 µM, immediately diluted to the indicated concentrations and incubated for 1 h at 4˚C (C) or RT (D). The OD at 400 nm was then measured. Bars show averages of values measured in three independent experiments performed under the same conditions and error bars show standard deviations.

## Longer polyQ sequences decrease nuclear localization of AR and increase DHT-dependent formation of AR-containing puncta in the nucleus

Transcriptional activation by AR requires the transport of AR to the nucleus, which is normally induced by ligand binding and exposure of a hidden NLS [18]. However, C-terminally truncated AR variants such as AR-V7 can be constitutively localized to the nucleus [20]. To analyze how the length of the polyQ sequence influences the nuclear localization of AR, we transfected LNCaP cells with Cherry-labeled AR variants containing 12, 20, 31 and 49 glutamines, and analyzed their distribution in the cytoplasm and the nucleus. The results of these

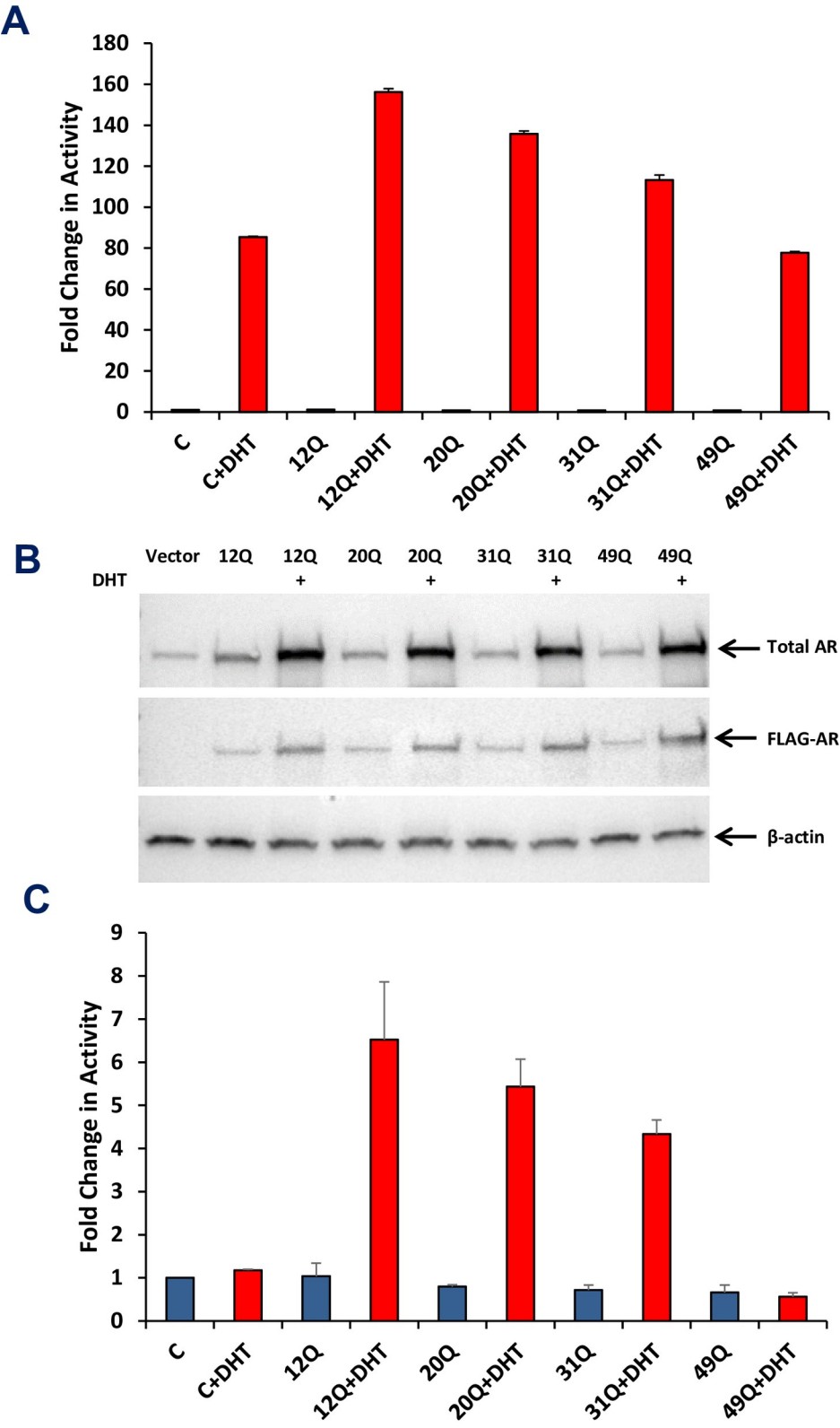

**Fig 3. Longer polyQ sequences decrease the transcriptional activity of AR.** (A,C) Luciferase reporter assays were performed with LNCaP cells and PC3 cells transfected with vectors expressing full-length AR containing 12, 20, 31 or 49 glutamines in the polyQ repeat. Luciferase activity was measured in the absence and presence of DHT, and the

activities were normalized to co-transfected β-galactosidase activity. Bars show averages of values measured in three independent experiments performed under the same conditions and error bars show standard deviations. (B) Western blot showing the levels of AR in the LNCaP cells used for the assays shown in panel (A). Top: total AR using an anti-AR antibody; middle: overexpressed FLAG-AR using an anti-Flag antibody; and bottom: anti-β-actin used as a loading control.

experiments need to be examined with caution, as we observed nuclear localization of the transfected proteins even in the absence of hormone (Fig 4A and 4B), which may arise because fluorescent proteins have a tendency to translocate to the nucleus [46]. Nevertheless, these experiments can still provide information regarding how the length of the polyQ sequence affects the propensity of AR for nuclear localization. Importantly, we observed that, under castration-mimicking conditions lacking hormone, longer polyQ sequences clearly favored the cytoplasmic localization of AR in detriment to its nuclear localization (Fig 4A and 4B). This finding did not arise from the differences in expression, as the levels of the expressed cherry-AR proteins assessed by Western blot exhibited some variability (Fig 4C) but this variability did not correlate with the percentage of protein localized to the nucleus. In the presence of DHT, most of the AR was localized to the nucleus regardless of polyQ length (Fig 5A). However, we observed that the nuclei contained puncta with higher Cherry fluorescence than the diffuse background, and that longer polyQ sequences led to more, brighter and larger puncta (Fig 5A). This conclusion was confirmed by measuring the ratio of fluorescence intensity in puncta to that of the diffuse background, which showed that the ratio correlated with polyQ length (Fig 5B). These results suggest that there are at least two potential mechanisms by which the increased tendency of AR to self-associate as polyQ length increases may decrease its transcriptional activity, i.e. a decreased tendency to translocate to the nucleus and an increased tendency to form nuclear puncta that might sequester AR and thus limit its access to the transcriptional machinery.

## Discussion

AR plays a key role in the development of CRPC. Understanding the mechanisms that control the transcriptional activity of AR and that underlie the emergence of androgen-independent AR activity is thus critical to design effective therapies to treat this devastating disease. The presence in AR of a large, unstructured NTD rich in low complexity sequences, together with growing evidence that this type of sequence often mediates incorporation of proteins into phase-separated membraneless organelles [39], suggest that one function of the AR NTD may be to mediate phase separation or at least to enhance the localization of AR into biomolecular condensates, thus modulating AR availability and/or activity. The study presented here now shows that low-complexity regions of both AR and KDM4A are involved in interactions between the two proteins in vitro and hence might contribute to epigenetic regulation of AR activity. Moreover, we find that the length of the polyQ sequence correlates with the tendencies of AR-NTD to phase separate in vitro, to remain in the cytoplasm and to form puncta inside the nucleus in the presence of DHT. Altogether, these findings lead us to propose the hypothesis that polyQ-dependent phase separation of AR may provide a means of negatively regulating its transcriptional activity. However, other explanations of our data are possible, and further studies will be needed to establish whether the nuclear foci observed here indeed form by liquid-liquid phase separation in cells, and to learn whether and how focus formation may be related mechanistically to effects on AR-mediated transcription.

Early work showed that the NTD is largely unstructured [31], but some of its sequences exhibit clear propensities to adopt secondary structure [22]. In particular, the polyQ sequence

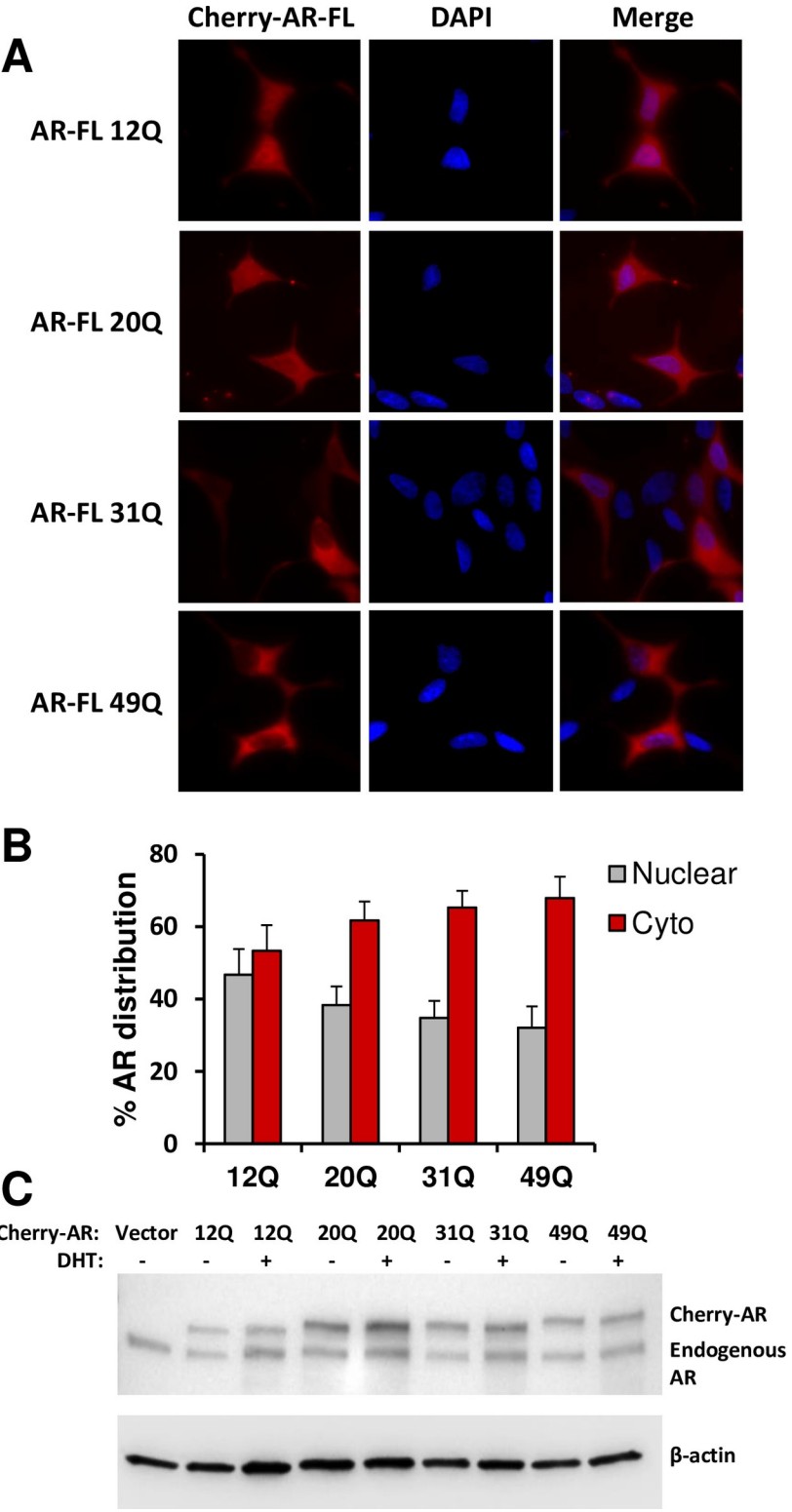

**Fig 4. Longer polyQ sequences decrease the nuclear localization of AR in the absence of hormone.** (A) Representative images of LNCaP cells transiently transfected with Cherry-AR containing different lengths of glutamine repeat (12Q, 20Q, 31Q or 49Q), in the absence of DHT. Red corresponds to Cherry fluorescence and blue to DAPI fluorescence. (B) AR distribution in the nucleus (grey bars) or the cytoplasm (red bars) normalized to the total in cells transiently transfected with Cherry-AR-12Q, -20Q, -31Q or -49Q. More than 50 cells were analyzed for each

condition. Bars show averages and error bars show standard deviations. (C) Western blot showing the expression levels of Cherry-AR and endogenous AR proteins using an anti-AR antibody and anti-β-actin antibody as a loading control.

of the AR NTD has an intrinsic tendency to form α-helical structure that increases in stability with the length of the sequence [34–36]. A leucine-rich region preceding the polyQ sequence also increases its propensity to form α-helical structure and impairs aggregation [36].

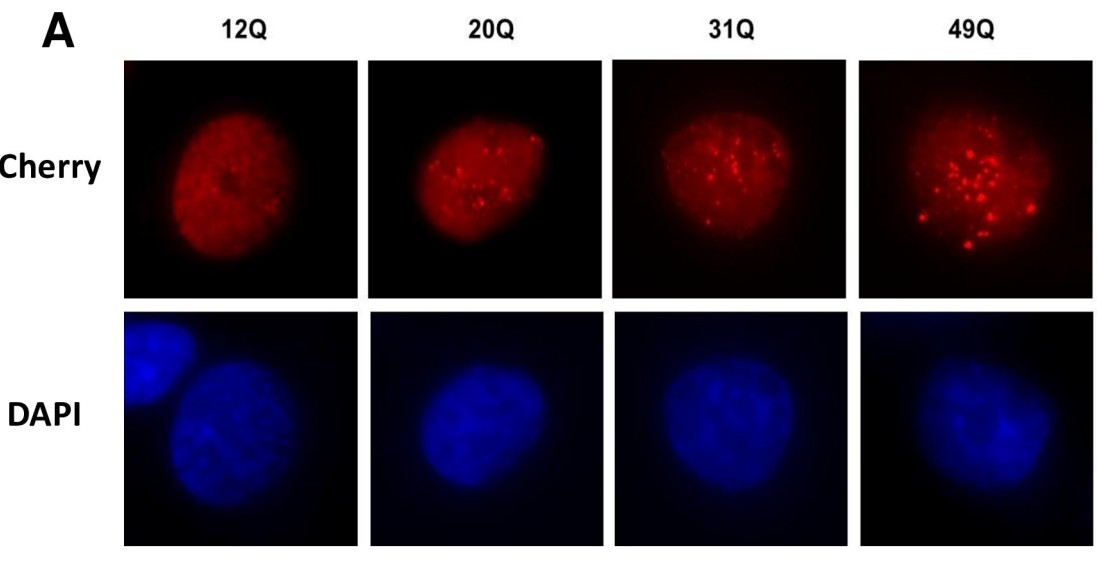

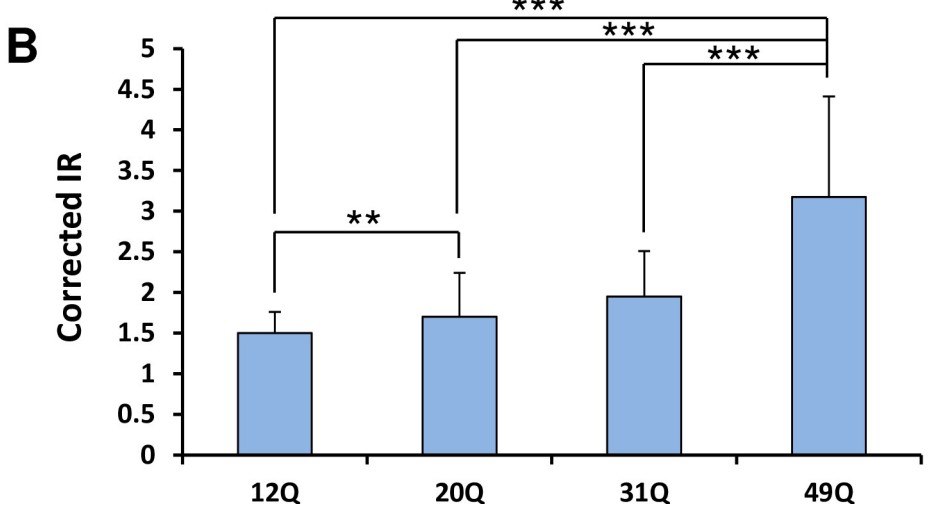

**Fig 5. Longer polyQ sequences increase the tendency of AR to localize to nuclear puncta in the presence of DHT.** (A) Representative images of cells stably expressing Cherry-AR-12Q, -20Q, -31Q or -49Q, treated with DHT 24 h before fixation. Red corresponds to Cherry fluorescence and blue to DAPI fluorescence. (B) Puncta intensity ratios (IRs) in cells stably expressing Cherry-AR-12Q, -20Q, -31Q or -49Q were calculated as the ratio of fluorescent intensity in the puncta minus the background divided by fluorescent intensity in the nucleoplasm minus the background. The background intensity corresponds to an image taken from an area where there were no cells. Bars represent average IRs calculate from measurements performed in at least 50 cells under each condition (5–45 puncta per cell) and errors bars show standard deviations. Statistical significance and P values were determined by one-way analysis of variance (ANOVA) with Holm-Sidak test (** $P < 0.01$; *** $P < 0.001$).

Conversely, exceedingly long polyQ sequences in AR favor aggregation that most likely involves beta-amyloid structure and can lead to formation of toxic fibrils, causing muscular atrophy [24, 25]. Since longer polyQ sequences protect against prostate cancer [26, 27] and lead to lower transcriptional activity [28–30] (Fig 3), it appears that the optimal length of the polyQ sequence may be the result of a trade-off between preventing over-activation of AR (if polyQ is too short) and minimizing the possibility of aggregation that leads to neurodegeneration (if polyQ is too long) [35, 47]. The increased stability of the helical structure formed by longer polyQ sequences was proposed to influence aggregation through formation of helical oligomers and to affect the transcriptional activity of AR by altering the strength of protein-protein interactions [35]. Our finding that the length of the polyQ sequence correlates with the tendency of the AR NTD to phase separate (Fig 2) shows that longer polyQ sequences increase self-association. However, it is unclear whether such self-association involves direct formation of oligomers with beta-structure or oligomerization via helix-helix interactions that may or may not convert to beta-amyloid-like structures. It is even plausible that there are multiple pathways to aggregation and the different pathways have different biological consequences (see below). Regardless of these possibilities, the observation that addition of 1,6-hexanediol reverses gel formation (Fig 2B) indicates that the self-association of the AR-NTD is reversible and therefore does not involve the irreversible formation of amyloid-like aggregates.

The correlation between polyQ length and propensity of the AR-NTD to phase separate (Fig 2), together with the inverse correlation between polyQ length and transcriptional activity [28–30] (Fig 3), suggest that polyQ-dependent phase separation of AR could provide a mechanism to reduce its activity. However, it is still unclear how the activity is reduced, and whether reduction directly arises from increased self-association. Formation of condensates can have a variety of effects on the processes contained within them [48]. By mass action alone, they can accelerate reactions through concentrating their components. This effect is likely to be most pronounced for reaction cascades, such as transcription, where multiple components must function together to produce a functional outcome. Condensates can also act beyond mass action to control biochemistry, for example by changing the kinetics of molecular interactions [49, 50], or the structural organization of enzymes and substrates [51], or the conformation of enzymes [52]. However, phase separation can also decrease biological activity by sequestering a protein and thus limiting access to its site of action, or by actively inhibiting it [48]. Our analyses of nuclear versus cytoplasmic localization lead us to two hypotheses that are not necessarily exclusive of each other. One hypothesis is based on the observation that longer polyQ sequences hinder the nuclear localization of AR in the absence of hormone (Fig 4), and postulates that longer polyQ sequences enhance the formation of AR oligomers in the cytoplasm, hindering access to the nuclear import machinery. Note that, although we did not observe formation of puncta containing AR in the cytoplasm, oligomers of limited sizes would not be distinguishable in our experiments. The second hypothesis arises from the observation that longer polyQ sequences led to increased formation of puncta in the nucleus when DHT was present (Fig 5), and proposes that localization of AR to these puncta sequesters the protein and hence prevents its incorporation into transcriptional complexes. One prediction of this hypothesis is that the AR foci do not correspond to AR-responsive genes, and that localization of AR to its proper loci is decreased with increasing polyQ length.

It is important to realize that other mechanisms may also underlie the reduction of AR activity caused by long polyQ sequences, such as increased affinity for transcriptional repressors. Moreover, it is unclear whether AR can undergo phase separation by itself in a cellular environment, as high concentrations of the AR-NTD are required for phase separation [37]. It is also plausible that the AR NTD facilitates recruitment to membraneless organelles that are formed through phase separation of other proteins. Indeed, the AR NTD was readily

incorporated in phase separated liquid droplets formed by speckle-type POZ protein (SPOP), an adaptor for a ubiquitin ligase that helps to recruit substrates for proteasomal degradation [37]. Thus, increased recruitment of AR to subcellular compartments for protein degradation might also underlie a decrease in AR activity with increased polyQ length. An additional possibility is that the AR NTD mediates differential incorporation into more than one type of membraneless organelle based on its length. For instance, AR might be recruited to organelles where it is sequestered or degraded and to other organelles where transcription of AR-dependent genes is activated, and where interactions between AR and KDM4s help recruiting either one of the two proteins to the condensate. Longer polyQ sequences may tilt the balance in favor of recruitment to the former type of condensate. Intriguingly, overexpression of the AR NTD was reported to delay progression of prostate cancer tumors and CRPC [53, 54]. The rationale behind these experiments was to use the AR NTD as a decoy molecule that competes with endogenous AR for binding to proteins required for its activation, and no interaction between the overexpressed AR NTD and endogenous AR was observed by co-immunoprecipitation. However, it is plausible that weak interactions underlying phase separation might not be detected by this method and that the decoy molecules helped to sequester endogenous AR through phase separation. If this notion is correct, overexpression of AR NTD containing a moderately long polyQ sequence (e.g. with 31 Qs) might provide a more effective means to delay progression of CRPC.

Clearly, there are many uncertainties about the ideas discussed above and much needs to be learned to understand how AR becomes activated in an androgen-independent manner in CRPC. Our results bring new hypotheses to this area that need to be tested with further research and that could open new therapeutic avenues for prostate cancer and muscular atrophy. It will also be important to investigate whether other nuclear receptors that also contain intrinsically disordered N-terminal domains, such as the estrogen receptor [55], also undergo phase separation, what are the functional consequences of such activity and whether phase separation is involved in diseases associated with these receptors.

## Materials and methods

### Protein expression and purification

Standard recombinant DNA techniques were used to prepare pGEX-KG or pET28 expression vectors encoding the human AR and human KDM4 fragments used in this study starting from vectors encoding the full-length proteins. His$_6$-AR-NTD fragments (aa 1–559) containing 12Q, 20Q, 31Q or 49Q were expressed with pET28 vectors at 25°C in *E. coli* BL21 (DE3) for 20 h with 0.5 mM isopropyl β-D-1-thiogalactopyranoside (IPTG). Cells were resuspended in buffer A (50 mM Tris pH7.5, 250 mM NaCl, 10 mM Imidazole, 6 M Guanidinium HCl, protease inhibitors and 10 mM 2-Mercaptoethanol) and frozen in liquid N2. Cells were thawed and disrupted using an Avestin cell disruptor and centrifuged at 20K, the supernatant was treated with protamine sulfate for 1 h at 4°C and centrifuged again. The supernatant was loaded onto NiNTA beads and incubated for 2 h at 4°C. Beads were later washed with 10 column volumes (CVs) of buffer A, 10 CVs buffer A containing 1M NaCl, 5 CVs buffer A and 5 CVs buffer B (50 mM Tris pH 7.5, 250 mM NaCl, 10 mM Imidazole, 6 M Urea and 10 mM 2-Mercaptoethanol). Proteins were eluted with buffer B containing 150 mM Imidazole, and subjected to anion exchange on a HiTrap Q column in 20mM Tris pH7.4 and 6M Urea. Purified proteins were concentrated and diluted 10x in buffer C (20 mM Tris pH7.5, 200 mM NaCl, 1 mM TCEP and 0.1 mM PMSF), concentrated again and dialyzed overnight at 4°C in buffer C.

His$_6$-AR-DBD-LBD (aa 554–919) was expressed with a pET28 vector at 18°C in *E. coli* BL21 (DE3) for 20 h with 0.2 mM isopropyl IPTG, with 10 μM of DHT and 10 μM ZnCl$_2$

added to the culture media. Cells were re-suspended in buffer A (20 mM Tris-HCl pH 7.5, 0.2 M NaCl, 10% glycerol, 10 μM zinc acetate, and 10 μM DHT) and lysed using an Avestin cell disruptor, centrifuged at 20K and the supernatant incubated with Ni-NTA beads for 2h at 4°C. The resin was washed with 500 ml of wash buffer (20 mM Tris-HCl pH 7.5, 0.2 M NaCl, 10% glycerol, 10 μM zinc acetate, 10 μM DHT, and 20 mM imidazole), treated with 20U/ml of Benzonase for 1 h at RT and then eluted with wash buffer containing 250 mM imidazole. TEV protease was added to the eluted protein and dialyzed overnight against 20 mM Tris-HCl pH 7.5, 150 mM NaCl, 10% glycerol, 10 μM $ZnCl_2$ and 10 μM DHT to remove the His6-tag. The protein was further purified using a Hiload 16/60 Superdex S200 gel filtration column in 20 mM Tris-HCl, pH 7.5, 150mM NaCl, 5% glycerol, 1 mM TCEP and 10 μM DHT.

Glutathione-S-transferase (GST)-AR-LBD (aa 663–919) was expressed with a pGEX-KG vector at 15°C in *E. coli* BL21 (DE3) for 18–24 h with 0.06 mM IPTG in LB media with 10 μM DHT. Cells were re-suspended in 50 mM Tris pH 7.2, 150 mM NaCl, 2 mM DTT, 10% Glycerol, protease inhibitor cocktail, 5 mM EDTA and 10 μM DHT. Cells were lysed on an Avestin cell disruptor and spun at 20K and incubated on a glutathione agarose-resin at 4°C overnight. The resin was washed with 5 CVs washing buffer (50 mM Tris, pH 8.0, 150 mM NaCl, 5 mM EDTA, 10% Glycerol, 10 μM DHT, 1 mM DTT), washed with 5 CVs benzonase buffer and incubated with 20U/ml of Benzonase at RT for 1–2 h. The resin was washed with TCB (50mM Tris, pH 8.0, 150 mM NaCl, 2.5 mM $CaCl_2$, 10 μM DHT, 10% Glycerol, 0.1% beta-Octyl Glucoside). Eluted protein was further purified by gel filtration on a Hiload16/60 Superdex S200 column in 25 mM Tris pH 7.4, 125 mM NaCl, 1 mM TCEP, 10 μM DHT, 5% glycerol, 0.1% beta-Octyl glucoside.

His$_6$-AR-NTD-DBD (aa 1–632) was expressed with a pET28 vector at 18°C in *E. coli* BL21 (DE3) for 20 h with 0.2 mM IPTG. Cells were re-suspended in buffer A (50 mM Tris pH7.5, 250 mM NaCl, 10 mM Imidazole, protease inhibitors and 10 mM 2-Mercaptoethanol) and frozen in liquid N2. Cells were thawed and disrupted using an Avestin cell disruptor and centrifuged at 20K and supernatant incubated with Ni-NTA beads for 2 h at 4°C. Beads were washed with 20 CVs of re-suspension buffer and treated with 20U/ml of Benzonase for 1 h at RT with rotation. Protein was eluted with re-suspension buffer containing 250 mM imidazole and subjected to Gel-filtration chromatography on a Superdex S200 column in 50 mM Tris, 150 mM NaCl and 1 mM TCEP.

GST-KDM4A-CD (aa 1–350), GST-KG-KDM4B-CD (aa 1–348) and GST-KG-KDM4C-CD (aa 1–350) were expressed with pGEX-KG vectors at 18°C in *E. coli* BL21 (DE3) for 20 h with 0.2 mM IPTG in LB media with 0.1 mM $ZnCl_2$ and 0.1 mM $FeSO_4$. Cells were re-suspended in 50 mM Tris pH 7.2, 300 mM NaCl, 2 mM DTT and protease inhibitors. After fast freezing in liquid N2, cells were broken on an Avestin cell disruptor, spun at 20K and loaded on glutathione agarose-resin at 4°C overnight. The resin was extensively washed with 10 CVs phosphate buffer saline (PBS), 10 CVs PBS containing 1 M NaCl, 5 CVs PBS and 5 CVs Benzonase buffer (25 mM Tris pH 8.0, 50 mM NaCl, 2 mM $MgCl_2$), and treated with 20 U/ml of Benzonase for 1 h at RT with slow rotation. The resin was then washed with 10 CVs PBS containing 1M NaCl, 10 CVs PBS and 5 CVs of thrombin cleavage buffer, then incubated with thrombin to remove the GST-tag for 3 h at RT. The protein was purified by Cation exchange chromatography on a HiTrap S column with 50 mM MES buffer pH6.5, followed by size exclusion chromatography on a Hiload 16/60 Superdex S75 column in 25 mM Tris pH 7.4, 125 mM NaCl and 1 mM TCEP.

His$_6$-KDM4A(301–708) was expressed with a pET28 vector at 18°C in *E. coli* BL21 (DE3) for 20 h with 0.5 mM isopropyl β-D-1-thiogalactopyranoside (IPTG). Cells were re-suspended in buffer A (50 mM Tris pH 7.5, 250 mM NaCl, 10 mM Imidazole, 6 M Guanidinium HCl, protease inhibitors and 10 mM 2-Mercaptoethanol). Cells were disrupted and the supernatant

was loaded onto NiNTA beads and washed with 10 CVs buffer A, 10 CVs buffer A containing 1M NaCl, 5 CVs buffer A and 5 CVs buffer B (50 mM Tris pH 7.5, 250 mM NaCl, 10 mM Imidazole, 6 M Urea and 10 mM 2-Mercaptoethanol). Proteins were eluted with buffer B containing 150 mM Imidazole and further purified by gel filtration on a S75 column in 20 mM Tris pH7.5, 200 mM NaCl, 1 mM TCEP.

His$_6$-KDM4A(703–1064) was expressed with a pET28 vector at 16˚C in *E. coli* BL21 (DE3) for 20 h with 0.5 mM IPTG in LB media and pellet re-suspended in buffer RB (50mM Tris pH = 7.5, 250 mM NaCl 4 mM Imidazole, protease inhibitors). After protein binding, the Ni-NTA beads were washed with RB and treated with Benzonase in 10 ml Benzonase buffer for 2h RT, washed again with RB and eluted with elution buffer (50 mM Tris pH 7.5, 250 mM NaCl 250–500 mM Imidazole). Proteins were then subjected to exclusion chromatography on a S200 column in 25 mM Tris pH 7.0, 125 mM NaCl and 1 mM TCEP.

GST-AR (93–495) was expressed with a pGEX-KG vector at 16˚C in *E. coli* BL21 (DE3) for 20 h with 0.5 mM IPTG in LB media. Cells were re-suspended in PBS, protease inhibitors cocktail and 5 mM DTT. Cells were broken on an Avestin cell disruptor, spun at 20K and incubated with glutathione agarose-resin at 4˚C overnight. The resin was extensively washed with 10 CVs PBS, 10 CVs PBS containing 1 M NaCl, 5 CVs PBS and 5 CVs Benzonase buffer (25mM Tris pH8.0, 50mM NaCl, 2 mM MgCl2), and treated with 20U/ml of Benzonase for 1–2 h at RT with rotation. The beads were washed with 5 CVs PBS and treated with 20U/ml of TEV protease to remove the GST tag overnight at 4˚C. Collected fractions were subjected to anion exchange chromatography on a HiTrap-Q column with 20mM Bis-Tris buffer pH 6.4 and size exclusion chromatography on a S75 column in 25 mM Tris pH 7.4, 125 mM NaCl and 2 mM TCEP.

To generate uniformly $^{15}$N-labeled proteins for NMR studies, bacteria were grown in M9 minimal medium supplemented with $^{15}$NH$_4$Cl (CIL, Andover, MA) as the sole nitrogen source. Proteins were purified as described above.

## NMR spectroscopy

All NMR spectra were acquired at 25˚C on a Varian INOVA 600 MHz spectrometer. Samples for $^1$H-$^{15}$N TROSY-HSQC measurements contained 40–50 μM of the $^{15}$N labelled protein and unlabeled protein. Experiments with $^{15}$N-KDM4A(1–350) or $^{15}$N-KDM4B(1–348) were performed in 25 mM Tris (pH 7.4), 125 mM NaCl and 1 mM TCEP with 5% D$_2$O. For samples containing $^{15}$N-AR-LBD or $^{15}$N-KDM4C(1–350), the buffer was 25 mM Tris (pH 7.4), 125 mM NaCl, 0.1% Octyl β-D-glucopyranoside and 1 mM TCEP with 5% D2O. For the NMR experiments including AR-LBD or AR-DBD+LBD, 1 μM DHT was included in the buffer. Total acquisition times were 2–10 h. NMR data were processed with NMRPipe [56] and analyzed with NMRView [57].

## Gel formation and turbidity assays

AR-NTD proteins containing 12Q, 20Q, 31Q or 49Qs at ~50uM concentration were dialyzed against a gelation buffer containing 20 mM Tris-HCl pH 7.5, 200 mM NaCl, 20 mM BME, 0.5 mM EDTA and 0.1 mM PMSF overnight. The proteins were then concentrated to 500 μM and incubated at RT for 24–48 h until gel formation was observed. For turbidity assays, protein samples in 25 mM Tris pH 7.5, 150 mM NaCl and 1 mM TCEP were concentrated to 100 μM and then diluted in the same buffer to the various concentrations shown in Fig 2C and 2D. The proteins were incubated at 4˚C or RT for 1 h, and the OD at 400 nm was then measured.

## Luciferase assay

LNCaP and PC3 cells were co-transfected with an ARE-luc construct containing three androgen response elements (ARE) ligated in tandem to a luciferase reporter [58], along with a control vector pcDNA3.1-FLAG or an AR-expression vector (pcDNA3.1-FLAG-AR) containing different polyQ sequence in the presence or absence of DHT. LNCaP cells were maintained in RPMI phenol red free with 5% charcoal stripped FBS during the assay. Transfections were carried out using FuGENE HD (Promega, Catalog. E2311) for LNCaP cells and Lipofectamine 3000 Kit for PC3 cells (Thermo Fisher, Catalog. L30000) following the instructions from the manufacturer. Cells were treated with vehicle or 10 nM Dihydrotestosterone (DHT) 24 h after transfection. Cell extracts were prepared 48 h after transfection and assayed for luciferase activity using the Promega luciferase detection kit. Luciferase activities were normalized to co-transfected β-galactosidase activity.

## Western blot analysis

Protein samples from LNCaP cells used to measure luciferase activity (described in Fig 3A) were loaded into 4% to 15% SDS-PAGE (Bio-Rad) and subjected to electrophoretic analysis and subsequent blotting. Nitrocellulose membranes were incubated with the primary antibody (overnight at 4˚C) and the relevant secondary antibodies (1 h at room temperature). The antibodies were purchased from MilliporeSigma: anti-AR (catalog 06–680); anti-FLAG M2 (catalog F3165) and anti-β-actin (catalog no. A5441). For the experiments shown in Fig 4C, LNCaP cells transiently transfected with Cherry-AR containing different polyQ lengths were blotted against the same anti-AR and anti-β-actin antibodies.

## AR localization assessment

LNCaP cells transiently transfected with Cherry-AR (full-length) containing different lengths of glutamine repeat (12Q, 20Q, 31Q or 49Q) were seeded onto coverslips in RPMI-1640 with 10% Charcoal-stripped fetal bovine serum (FBS), and continued to grow for 48 h under the same starvation conditions. Coverslips were fixed with methanol for 10 min at -20˚C, washed with PBS and mounted on slides with mounting media containing 4′,6-diamidino-2-phenylindole (DAPI). Cells were imaged as z-stacks using a widefield Delta Vision Fluorescence microscope. Maximum projections were done before nuclear/cytoplasmic localization of cherry-AR-FL was quantified using the ImageJ Macro "Intensity Ratio Nuclei Cytoplasm Tool". The data were plotted as the percentage of nuclear and cytoplasmic distribution of Cherry AR. More than 50 cells were analyzed per experimental condition.

## Puncta formation assay

LNCaP cells stably expressing Cherry-AR containing different lengths of glutamine repeat (12Q, 20Q, 31Q or 49Q) were seeded onto coverslips in RPMI-1640 with 10% Charcoal stripped FBS and continued to grow for 48 h under the same starvation conditions. Twenty four h before fixation, the cells were treated with 10 nM DHT to induce AR nuclear localization. Coverslips were fixed with methanol for 10 min at -20˚C and mounted on coverslips with mounting solution containing 4′,6-diamidino-2-phenylindole (DAPI). A widefield Delta Vision Fluorescence microscope was used to take z-stacks images through whole cells, maximum projections were performed, and the puncta intensity of cherry-AR-FL was quantified using ImageJ. More than 50 cells were analyzed per experiment per condition, and there were ~5–45 puncta per cell. Intensity ratio (IR) was calculated as the ratio of fluorescent intensity in the puncta minus the background divided by the fluorescent intensity in the nucleoplasm

minus the background. The background intensity was obtained from an image of an area where there were no cells.

## Supporting information

**S1 Fig. Disorder prediction in AR and KDM4A. A.** Amino acid sequence of the human androgen receptor N-terminal domain. The poly-Q is highlighted in yellow. **B, C.** Intrinsically disordered scores predicted for human AR (**B**) and human KDM4A (**C**) by the disordered prediction tool IUPred3 [Erdos G, Pajkos M, Dosztanyi Z. IUPred3: prediction of protein disorder enhanced with unambiguous experimental annotation and visualization of evolutionary conservation. Nucleic Acids Res. 2021;49(W1):W297-W303].
(PDF)

**S2 Fig. The AR NTD also binds to the catalytic domains of KDM4B and KDM4C. A.** Model of how AR interacts with KDM4A. The model predicts that binding involves interactions of the AR NTD and DBD domains with the catalytic domain of KDM4A, as well as of the AR LBD with the region of KDM4A spanning residues 301–708. Intramolecular interactions between the DBD and LBD domains are postulated to hinder the DBD/KD4MA interactions. **B.** The diagrams show superpositions of $^1$H-$^{15}$N TROSY-HSQC spectra of $^{15}$N-KDM4B(1–348) (left) or $^{15}$N-KDM4C(1–350) (right) alone (black contours) or in the presence of equimolar amounts of unlabeled AR-NTD-DBD (red).
(PDF)

**S3 Fig. SDS PAGE analysis of samples used in the turbidity assays of Fig 2.** The samples were separated by SDS PAGE followed by coomassie blue staining. Molecular weight markers are on the left of each gel.
(PDF)

**S1 Raw images. Uncropped gel images.**
(PDF)

## Acknowledgments

We thank Yilun Sun for technical assistance.

## Author Contributions

**Conceptualization:** Carlos M. Roggero, Michael K. Rosen, Zhi-Ping Liu, Josep Rizo.

**Data curation:** Carlos M. Roggero, Victoria Esser, Josep Rizo.

**Formal analysis:** Carlos M. Roggero, Victoria Esser, Lingling Duan, Allyson M. Rice, Ganesh V. Raj, Michael K. Rosen, Zhi-Ping Liu, Josep Rizo.

**Funding acquisition:** Zhi-Ping Liu, Josep Rizo.

**Investigation:** Carlos M. Roggero, Victoria Esser, Shihong Ma, Zhi-Ping Liu, Josep Rizo.

**Methodology:** Carlos M. Roggero, Victoria Esser, Lingling Duan, Allyson M. Rice, Ganesh V. Raj, Michael K. Rosen, Zhi-Ping Liu, Josep Rizo.

**Project administration:** Josep Rizo.

**Supervision:** Zhi-Ping Liu, Josep Rizo.

**Validation:** Carlos M. Roggero, Josep Rizo.

**Writing – original draft:** Carlos M. Roggero, Josep Rizo.

**Writing – review & editing:** Victoria Esser, Michael K. Rosen, Zhi-Ping Liu.

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
