## [Decision Letter · Decision Letter 0]

11 Nov 2021

PONE-D-21-32225Poly-glutamine-dependent self-association as a potential mechanism for regulation of androgen receptor activityPLOS ONE

Dear Dr. Rizo,

Thank you for submitting your manuscript to PLOS ONE. After careful consideration, we feel that it has merit but does not fully meet PLOS ONE’s publication criteria as it currently stands. Therefore, we invite you to submit a revised version of the manuscript that addresses the points raised during the review process. Please include a point-by-point repose to all comments and the corresponding revisions made.

We look forward to receiving your revised manuscript.

Kind regards,

Bostjan Kobe, Ph.D.

Academic Editor

PLOS ONE

Journal Requirements:

"We thank Yilun Sun for technical assistance. This work was supported by grants from the Cancer Prevention and Research Institute of Texas (CPRIT) (RP120717-AC and RP120717-P1 to Z.P.L.; RP120717-P3 to J.R.), NIH (RO1 CA215063 to Z.P.L.), and from the Welch Foundation (I-1304 to J.R.)."

"J.R. RP120717-P3 Cancer Prevention and Research Institute of Texas (CPRIT)

https://www.cprit.state.tx.us/

J.R. I-1304  Welch Foundation

https://welch1.org/

Z.P.L. RP120717-AC and RP120717-P1 Cancer Prevention and Research Institute of Texas (CPRIT)

https://www.cprit.state.tx.us/

Z.P.L. RO1 CA215063 NIH

https://www.nih.gov/

The funders had no role in study design, data collection and analysis, decision to publish, or preparation of the manuscript"

Reviewers' comments:

Reviewer's Responses to Questions

**Comments to the Author**

1. Is the manuscript technically sound, and do the data support the conclusions?

Reviewer #1: Yes

Reviewer #2: Yes

2. Has the statistical analysis been performed appropriately and rigorously? 

Reviewer #1: Yes

Reviewer #2: Yes

3. Have the authors made all data underlying the findings in their manuscript fully available?

Reviewer #1: Yes

Reviewer #2: Yes

4. Is the manuscript presented in an intelligible fashion and written in standard English?

Reviewer #1: Yes

Reviewer #2: Yes

5. Review Comments to the Author

Reviewer #1: In this manuscript, Roggero et al. studied the interactions of various domains/regions of AR and KDMs using NMR spectroscopy and other biophysical techniques in order to determine the mechanism of interaction of such a critical epigenetic complex. The authors also studied LLPS properties and transcriptional activity effects of various polyQ variants of AR, demonstrating that longer polyQ favor phase separation in vitro and formation of AR puntae in the nucleus.

This work focuses on the interaction mechanism of a very important biological molecule (AR) and its interaction with equally important family of proteins (epigenetic modifiers). Not only is AR (and other nuclear hormone receptors) critically important in prostate (and other) cancers drivers, but also in neurodenerative diseases such as poly Q-mediated Spinobulbular muscular atrophy.

Due to the presence of multiple folded domains interspersed with long sticky IDRs, such proteins and their complexes are usually very difficult to study structurally and biophysically, because of the lack of inherent stability (i.e. degradation and precipitation/aggregation issues). The strategy and approaches taken by the authors are appropriate, and could potentially be the foundation for future studies of full-length AR. I think that this paper should be published after the following are addressed:

1. Because various domain(s) with and without IDRs are used in the binding studies, I think that the authors should provide a model of how they think the different domains are interacting within the complex. For example, to me, it seems that only the NTD (and not the LBD) of AR binds to the catalytic domain, but the LBD helps, as if it is relieving of some intra-molecular interaction NTD interactions.

2. Also, can thee authors comment on why they are saying that the interactions are weak. I think tight and weak interactions should only be based upon some form of titration expts, and not on degree of changes of intensity/broadening of a single concentration complex.

3. Can the authors provide the sequence (or at least part) of the NTD showing where these poly Q expansions are located? Also, can they provide the disorder predictions (or other structural/biophysical property) of the different KDMs used, or comment on the rationale why these different proteins interact with AR differently.

4. Personally, the evidence of interaction of the complex via NMR spectroscopy is more than enough to demonstrate binding. The cross-linking linking expts only complicate the studies as one also need to also show a negative control, where some IDR-containing constructs from at least one of the complex component did not show binding. In such dynamics systems, cross-linking could potentially be a problem, since, unlike in the case of fold domain-folded domain interaction, the chemical cross-linker can attach to some site (and changes the energy landscape) and ‘reach out’ to the binding binder because everything is just dynamic. So, a negative control is always needed, to demonstrate that such scenario doesn’t exist.

5. The authors should also mention that their studies will have impact on not only understanding the mechanism of AR-mediated cancer, but also on muscular atrophy.

Reviewer #2: In their manuscript entitled, “Poly-glutamine-dependent self-association as a potential mechanism for regulation of androgen receptor activity,” Rizo et al investigate the implications of various forms of androgen receptor on interactions with an epigenetic enzyme, transcriptional output, aggregation, and cellular localization. This is of importance as these types of changes alter prostate cancer progression but are not well understood. Specifically, the authors carried out the following experiments: a form of TROSY NMR to determine which domains and polyq NTD lengths affected binding with KDM4 (and isoforms), cross-linking of these as confirmation of the NMR, probed phase separation in vitro as a consequence of NTD AR polyq alteration, phase separation due to increasing poly q length was shown to be reversible (so as to be different from amyloid formation), transcriptional alteration (by means of transactivation assays) were used to show how polyq extensions altered transcription, and different polyq lengths on NTD of AR were monitored in the cell to show the degree of nuclear localization and the formation of punctate structures in the nucleus with varying degrees of polyq sequence. The experiments are of high quality and the scientific conclusions are well justified. It is also important to note that these experiments, due to the difficult nature of working with, purifying, and handling AR are not trivial and require much work. The paper is well written and easy to follow with very few errors. I recommend publication with just some very small changes:

-The word dramatic in the abstract could probably be changed to something more precise such as extensive

-In the abstract it might be good to define low-complexity sequences and why only polyQ is the one to look at here

-In the abstract, KDM4 is not introduced. It should be given a small introduction/rationale for study

-In the intro, paragraph starting line 62, it might be useful for the reader to have the term AF (activation function) introduced along with coregulators. Just a sentence or two so that the general reader knowns how and where and which type of coregulators are recruited to incite transcription.

-The term intrinsically disordered is not mentioned here regarding AF1. Is this because it has some structure? More detail on this especially in regard to other NRs (one example that is well studied for example is ER) and the intrinsically disordered nature and differences between receptors could be useful.

-It wasn’t clear what the mechanism of addition of polyq sequences in a cancer setting is? What is known about how poly q sequences are added to AR? A very small section in intro might help clear this up to the reader, even if it is unknown.

-Line 112 define what a Tudor domain is

-In the discussion it would be nice to see more discussion about the relationship, in general (not just with AR) about the relationship between in vitro phase separation and cellular effects. Are there references to this that could be explained? There may not be much out there on this subject.

-Page 12, line 259, the comma before however should probably be changed to a semicolon

6. PLOS authors have the option to publish the peer review history of their article (what does this mean?). If published, this will include your full peer review and any attached files.

Reviewer #1: No

Reviewer #2: **Yes: **John B. Bruning

---

## [Author Response · Author response to Decision Letter 0]

3 Dec 2021

Reviewers' comments:

5. Review Comments to the Author

Reviewer #1: In this manuscript, Roggero et al. studied the interactions of various domains/regions of AR and KDMs using NMR spectroscopy and other biophysical techniques in order to determine the mechanism of interaction of such a critical epigenetic complex. The authors also studied LLPS properties and transcriptional activity effects of various polyQ variants of AR, demonstrating that longer polyQ favor phase separation in vitro and formation of AR puntae in the nucleus.

This work focuses on the interaction mechanism of a very important biological molecule (AR) and its interaction with equally important family of proteins (epigenetic modifiers). Not only is AR (and other nuclear hormone receptors) critically important in prostate (and other) cancers drivers, but also in neurodenerative diseases such as poly Q-mediated Spinobulbular muscular atrophy.

Due to the presence of multiple folded domains interspersed with long sticky IDRs, such proteins and their complexes are usually very difficult to study structurally and biophysically, because of the lack of inherent stability (i.e. degradation and precipitation/aggregation issues). The strategy and approaches taken by the authors are appropriate, and could potentially be the foundation for future studies of full-length AR. I think that this paper should be published after the following are addressed:

We very much appreciate the positive evaluation of our paper and the constructive criticisms.

1. Because various domain(s) with and without IDRs are used in the binding studies, I think that the authors should provide a model of how they think the different domains are interacting within the complex. For example, to me, it seems that only the NTD (and not the LBD) of AR binds to the catalytic domain, but the LBD helps, as if it is relieving of some intra-molecular interaction NTD interactions.

We now present a model that summarizes the AR/KDM4A interactions in S2 FigA, and cite this figure in line 160.

2. Also, can thee authors comment on why they are saying that the interactions are weak. I think tight and weak interactions should only be based upon some form of titration expts, and not on degree of changes of intensity/broadening of a single concentration complex.

As we explain in lines 137-141, we performed extensive pulldown assays and were not able to consistently observe interactions between fragments from AR and KDM4A. These findings lead us to suggest that the interactions are weak. In the revised manuscript we now remind the reader of this point in the summary at the end of this section (line 175).

3. Can the authors provide the sequence (or at least part) of the NTD showing where these poly Q expansions are located? Also, can they provide the disorder predictions (or other structural/biophysical property) of the different KDMs used, or comment on the rationale why these different proteins interact with AR differently.

The sequence of the AR NTD, with the polyQ region highlighted, and the disorder predictions for AR and KDM4A are now shown in S1 Fig. We can only speculate that differences in interactions arise from differences in the sequences, but further work will be required to understand which are the determinants of specificity. 

4. Personally, the evidence of interaction of the complex via NMR spectroscopy is more than enough to demonstrate binding. The cross-linking linking expts only complicate the studies as one also need to also show a negative control, where some IDR-containing constructs from at least one of the complex component did not show binding. In such dynamics systems, cross-linking could potentially be a problem, since, unlike in the case of fold domain-folded domain interaction, the chemical cross-linker can attach to some site (and changes the energy landscape) and ‘reach out’ to the binding binder because everything is just dynamic. So, a negative control is always needed, to demonstrate that such scenario doesn’t exist.

We agree with the reviewer and have removed the cross-linking data. As a consequence, we have renumbered the subsequent figures as indicated above.

5. The authors should also mention that their studies will have impact on not only understanding the mechanism of AR-mediated cancer, but also on muscular atrophy.

We now mention the potential impact of our results for muscular atrophy in the abstract and in the last paragraph of the discussion (line 384). Moreover, in the introduction (line 95) we have replaced the term ‘neurodegenerative disease’ with ‘muscular atrophy’.

Reviewer #2: In their manuscript entitled, “Poly-glutamine-dependent self-association as a potential mechanism for regulation of androgen receptor activity,” Rizo et al investigate the implications of various forms of androgen receptor on interactions with an epigenetic enzyme, transcriptional output, aggregation, and cellular localization. This is of importance as these types of changes alter prostate cancer progression but are not well understood. Specifically, the authors carried out the following experiments: a form of TROSY NMR to determine which domains and polyq NTD lengths affected binding with KDM4 (and isoforms), cross-linking of these as confirmation of the NMR, probed phase separation in vitro as a consequence of NTD AR polyq alteration, phase separation due to increasing poly q length was shown to be reversible (so as to be different from amyloid formation), transcriptional alteration (by means of transactivation assays) were used to show how polyq extensions altered transcription, and different polyq lengths on NTD of AR were monitored in the cell to show the degree of nuclear localization and the formation of punctate structures in the nucleus with varying degrees of polyq sequence. The experiments are of high quality and the scientific conclusions are well justified. It is also important to note that these experiments, due to the difficult nature of working with, purifying, and handling AR are not trivial and require much work. The paper is well written and easy to follow with very few errors. I recommend publication with just some very small changes:

We also thank this reviewer for the positive evaluation of our manuscript and for the constructive criticisms.

-The word dramatic in the abstract could probably be changed to something more precise such as extensive

We have changed ‘dramatic’ to ‘extensive’

-In the abstract it might be good to define low-complexity sequences and why only polyQ is the one to look at here

Because of the limited space available in an abstract, we prefer to define the term low-complexity sequence in the introduction. We have now inserted such definition in lines 91-92.

In the abstract, the rationalize for focusing on the polyQ sequence is explained in the sentence ‘Longer polyQ sequences were reported to decrease transcriptional activity and to protect against prostate cancer’.

-In the abstract, KDM4 is not introduced. It should be given a small introduction/rationale for study

The rationale for studying KDMs is given in the sentence ‘Development of castration resistant prostate cancer … entails extensive epigenetic changes depending in part on histone lysine demethylases (KDMs) that interact with AR’

-In the intro, paragraph starting line 62, it might be useful for the reader to have the term AF (activation function) introduced along with coregulators. Just a sentence or two so that the general reader knowns how and where and which type of coregulators are recruited to incite transcription.

We have included the following sentence at the beginning of this paragraph (lines 88-90):

‘The AR NTD includes sequences that are collectively named activation function 1 (AF1) and are important for AR-dependent gene regulation, interacting with the general transcription factor TFIIF and co-activators such SRC-1, CBP and ART27 (reviewed in [22]).’

-The term intrinsically disordered is not mentioned here regarding AF1. Is this because it has some structure? More detail on this especially in regard to other NRs (one example that is well studied for example is ER) and the intrinsically disordered nature and differences between receptors could be useful.

We had not used the term intrinsically disordered, but we had mentioned that the AR NTD is largely unstructured albeit with some structural propensities in some sequences. Before mentioning these results, we now have inserted the following sentence (lines (100-101):

‘As expected because of the abundance of low complexity sequences, much of the AR NTD is predicted to be intrinsically disordered (S1B Fig)’

We have also added the following sentence at the end of the discussion (lines 384-387):

‘It will also be important to investigate whether other nuclear receptors that also contain intrinsically disordered N-terminal domains, such as the estrogen receptor [55], also undergo phase separation, what are the functional consequences of such activity and whether phase separation is involved in diseases associated with these receptors.’

-It wasn’t clear what the mechanism of addition of polyq sequences in a cancer setting is? What is known about how poly q sequences are added to AR? A very small section in intro might help clear this up to the reader, even if it is unknown.

In lines 93-94, we have inserted the following sentence:

‘This variability arises because of slippage of DNA polymerase on the DNA template containing multiple copies of the same codon during DNA replication [18]’

-Line 112 define what a Tudor domain is

In lines 132-133 we now mention that PHD and Tudor domains recognize methylated basic residues and cite a review on KDMs that discusses their sequences (reference 41).

-In the discussion it would be nice to see more discussion about the relationship, in general (not just with AR) about the relationship between in vitro phase separation and cellular effects. Are there references to this that could be explained? There may not be much out there on this subject.

We have included the following paragraph in the discussion (line 340-346):

‘Formation of condensates can have a variety of effects on the processes contained within them [48]. By mass action alone, they can accelerate reactions through concentrating their components. This effect is likely to be most pronounced for reaction cascades, such as transcription, where multiple components must function together to produce a functional outcome. Condensates can also act beyond mass action to control biochemistry, for example by changing the kinetics of molecular interactions [49, 50], or the structural organization of enzymes and substrates [51], or the conformation of enzymes [52].’

-Page 12, line 259, the comma before however should probably be changed to a semicolon

We have change the phrase a little to put the however at the beginning. The sentence now reads (lines 311-312):

‘However, other explanations of our data are possible, …

We hope that we have properly addressed all the concerns and that the paper is now acceptable for publication in its current form. We would like to thank again the reviewers for their valuable comments. Please do not hesitate to contact me if you have any questions.

---

## [Editor Report · Decision Letter 1]

14 Dec 2021

Poly-glutamine-dependent self-association as a potential mechanism for regulation of androgen receptor activity

PONE-D-21-32225R1

Dear Dr. Rizo,

We’re pleased to inform you that your manuscript has been judged scientifically suitable for publication and will be formally accepted for publication once it meets all outstanding technical requirements.

Kind regards,

Bostjan Kobe, Ph.D.

Academic Editor

PLOS ONE
---

## [Editor Report · Acceptance letter]

27 Dec 2021

PONE-D-21-32225R1 

Poly-glutamine-dependent self-association as a potential mechanism
for regulation of androgen receptor activity 

Dear Dr. Rizo:

I'm pleased to inform you that your manuscript has been deemed suitable for publication in PLOS ONE. Congratulations! Your manuscript is now with our production department. 

Kind regards, 

on behalf of

Professor Bostjan Kobe 

Academic Editor

PLOS ONE